# The Neuroscience of Growth Mindset and Intrinsic Motivation

**DOI:** 10.3390/brainsci8020020

**Published:** 2018-01-26

**Authors:** Betsy Ng

**Affiliations:** National Institute of Education, Nanyang Technological University, 1 Nanyang Walk, Singapore 637616, Singapore; betsy.ng@nie.edu.sg

**Keywords:** growth mindset, intrinsic motivation, education, neuroscience, neural

## Abstract

Our actions can be triggered by intentions, incentives or intrinsic values. Recent neuroscientific research has yielded some results about the growth mindset and intrinsic motivation. With the advances in neuroscience and motivational studies, there is a global need to utilize this information to inform educational practice and research. Yet, little is known about the neuroscientific interplay between growth mindset and intrinsic motivation. This paper attempts to draw on the theories of growth mindset and intrinsic motivation, together with contemporary ideas in neuroscience, outline the potential for neuroscientific research in education. It aims to shed light on the relationship between growth mindset and intrinsic motivation in terms of supporting a growth mindset to facilitate intrinsic motivation through neural responses. Recent empirical research from the educational neuroscience perspective that provides insights into the interplay between growth mindset and intrinsic motivation will also be discussed.

## 1. Introduction

With an emphasis on inquiry and scientific skills, students are encouraged to discover, produce and evaluate knowledge, using inquiry and scientific skills [1]. Such inquiry learning should be structured in a way that student learning is facilitated, while encouraging students to plan and conduct their own investigation. An autonomy-supportive environment facilitates autonomous learning, and fosters self-determined motivation in students [2]. Students learn to synthesize contradictory perspectives and rise to intellectual meta-levels of thinking, which is a crucial trait for the 21st-century operating environment [3]. As such, it is fundamental to nurture the young generation in becoming adaptive, self-regulated and self-determined.

In the 21st century, there has been a strong proliferation of research on growth mindset and intrinsic motivation in learning. The constructs of mindset and motivation have been important foci among educators seeking to positively impact student learning and outcomes. The underlying mechanism for students to have their own agency in finding out new knowledge is intrinsic motivation. However, much of this research has relied on quantitative approaches for assessing students’ self-reports on motivational regulations and learning outcomes [4,5]. Some of these quantitative findings are used to generalize across school settings. Although the multiple facets of student motivation and learning have been identified in quantitative analyses, they have not provided a detailed understanding of students’ motivational processes. Neuroscience methods may offer new insights regarding students’ motivation and learning processes. 

Most neuroscience studies have focused on research related to cognitive functions, such as attention, memory and decision-making. In addition to these cognitive studies, there is also the implicit nature of mindsets that lead to the malleability of self-attributes (e.g., intelligence) [6]. Subtle feedback and messages related to growth mindset can have noticeable effects on students’ attitudes and motivation that may transfer to long-term outcomes. Likewise, human motivation is important, as it is one’s intrinsic desire to learn and obtain information. Growth mindset is the belief that intelligence can be nurtured through learning and effort, while intrinsic motivation is the volition to engage in a task for inherent satisfaction. Individuals with growth mindset believe that motivation can be nurtured, and that extrinsic motivation can be internalized (i.e., from extrinsic regulation to integrated regulation that is similar to intrinsically motivated behavior). In an integrative view, growth mindset and intrinsic motivation are important and interrelated, thus raising fundamental questions about the neural mechanisms of mindset-motivation interaction. The links among growth mindset, brain and motivation are important to academic performance. Therefore, it is important to draw on neuroscientific findings to show the way the brain is motivated, and how it learns by changing mindset (i.e., from a fixed to a growth mindset). Such intervention studies are still not common, and there is potential in these research areas.

This paper reviews the theoretical frameworks of growth mindset and intrinsic motivation, and how they are linked to neuroscientific evidence. It also reviews a number of recent neuroscience studies related to growth mindset and intrinsic motivation. It is important to survey the progress of neuroscience research on growth mindset and intrinsic motivation, as understanding the neural substrates will provide insights into human motivation and drive. Neuroscience research has the potential to support and refine models of motivation and cognitive skill. It may play a pivotal role in developing classroom interventions and understanding non-cognitive skills (e.g., mindset). Knowing the key brain regions that are associated with growth mindset and intrinsic motivation, researchers and practitioners could work together to investigate the granular processes of motivation in relation to growth mindset.

Most empirical research on growth mindset and intrinsic motivation has focused on behavioral methods and self-reports of experiences. There is little information about the internal processes of motivation at a higher level of resolution. It is, therefore, relevant and timely to examine the existing literature and empirical research that is associated with intrinsic motivation. Neuroscientific evidence has the potential to uncover new insights and refine the conceptual ideas of intrinsic motivation by articulating the granular processes of motivation that behavioral methods alone cannot afford. This paper offers recommendations for potential neuroscience research in studying growth mindset and intrinsic motivation.

## 2. Growth Mindset

Growth mindset is defined as a belief that construes intelligence as malleable and improvable [6]. Students with growth mindset are likely to learn by a mastery approach, embrace challenges and put in effort to learn. For instance, growth-minded individuals perceive task setbacks as a necessary part of the learning process and they “bounce back” by increasing their motivational effort [7,8]. One recent study on elementary students showed that leveraging an online educational game (the BrainPOP website) with in-game rewards can promote a growth mindset by directly incentivizing effort and encouraging persistence in low performing students [7]. Learners with growth mindset tend to embrace lifelong learning and the joy of incremental personal growth. In addition, they do not see their intelligence or personality as fixed traits. They will mobilize their learning resources without being defeated by the threat of failure. This paper aims to provide some insights into the cultivation of resilience and mastery in university students, preparing them to overcome challenges in the real working world.

Empirical studies have revealed that growth mindset has positive effects on student motivation and academic performance [9,10]. Recent research has also shown that mindset is related to student outcomes and behaviors including academic achievement, engagement, and willingness to attempt new challenges [11,12]. Numerous studies have shown the effects of growth mindset interventions on students’ achievement at all ages. According to Dweck [9], teaching growth mindset to junior high school students resulted in increased motivation and better academic achievement. Her findings revealed that students in the growth mindset intervention group outperformed those in the control group (who received excellent training in study skills), indicating improved learning and desire to work hard. The growth mindset intervention teaches students that intelligence is not a fixed quality [13]. Intelligence can be nurtured through challenging tasks, as intelligence grows with hard work on challenging problems. A growth mindset intervention was especially impactful with student outcomes in particular subjects such as science and mathematics [14]. 

An individual with a growth mindset works hard and improves without an incentive reward in mind as the outcome. The conceptualization of growth mindset is similar to that of intrinsic motivation. A learner with a growth mindset tends to self-regulate their own learning and has the propensity to cope with academic tasks. Hence, encouraging a growth mindset can improve the academic performance of college students [14,15] and middle school math students [9]. 

Most of the abovementioned empirical studies reported the utility of questionnaires or self-report measures. There is still limited neuroscientific research on the neural mechanism of growth mindset. It is, therefore, important to examine data from other means such as neuroscientific information about how the brain changes with experience of learning and how it is associated to growth mindset. The subsequent sections will discuss the neuroscientific evidence of growth mindset.

## 3. Intrinsic Motivation 

Intrinsic motivation is inherent, as it drives the direction of an individual’s behavior and self-determination [16]. Self-determination is important in the development of beings to become more effective and refined in their reflection of ongoing experiences [17]. When students experience the inherent satisfaction of the activity itself, they will show intrinsically motivated behavior. If students are doing the activity in order to attain some reward, such as grades or social recognition, they are extrinsically motivated [18]. Students’ motivated behaviors pertaining to choice, effort and persistence in academic tasks correspond directly with their level of intrinsic motivation [19,20]. 

Numerous studies have examined the effects of intrinsic motivation, including the adaptive consequences for individuals such as exposing them to novel situations and developing their diverse competencies to cope with unforeseen circumstances [21]. In addition, intrinsic motivation is the propensity for individuals to learn about new subjects and to differentiate their interests, thereby fostering a sense of purpose and meaning [22]. Recent empirical findings have shown that intrinsic motivation is a key factor in academic achievement [23] and pursuit of interest [24], thus fostering learning and growth. 

Dopamine is the predominant neurotransmitter in the brain that aids in controlling the brain’s reward and pleasure centers, as well as motivated and emotional behaviors [25]. Dopamine neurons that are excited by unexpected reward events project to the striatum, cortex, limbic system and hypothalamus, thus affecting physiological functions and motivated behaviors. Dopamine is considered a key substrate of intrinsic motivation, thus promoting attentiveness and behavioral engagement [25]. For instance, participants were likely to voluntarily engage with the task during a free-choice time period [26] or a self-determined choice condition [27]. These consistent findings indicate that an enhanced activity within the dopaminergic value system whereby perceived autonomy support promotes intrinsic motivation. As such, learning is a neural process that requires the reinforcement of synaptic functioning and is strongly mediated by dopamine and attentional gain in the frontal cortex [28]. Positive and negative affect will also strengthen or weaken the learner’s intrinsic motivation in a particular subject, thus influencing the attitude towards that subject. 

Over the past few decades, behavioral evidence has established the importance of intrinsic motivation and how it impacts one’s learning. However, our understanding of the underlying mechanism of intrinsic motivation is still in its infancy, and it is unclear how one’s intrinsic motivation progresses or changes over time. More evidence is needed to establish the mechanism of intrinsic motivation at a granular level. The recommendation is to include neuroscientific evidence to track and understand which aspects of one’s learning progress determine intrinsic motivation, complementing the existing behavioral evidence. An approach of the neuroscience method is to foster intrinsically motivated behaviors based on task complexity in various contexts, thereby addressing intrinsic motivation through different forms of exploration. The following sections will discuss in detail the neuroscience methods and neuroscientific evidence of intrinsic motivation.

## 4. Neuroscience Methods

The main neuroscience methods that have been applied in motivation studies are electroencephalography (EEG) and functional magnetic resonance imaging (fMRI). Such neuroscientific research is still considered novel, as most motivational studies have focused on behavioral methods. Both neuroscientific techniques are non-invasive procedures for measuring brain activity. The key difference is fMRI has a higher spatial resolution than EEG, whereas EEG has a better temporal resolution than fMRI. 

Neuroscience methods (e.g., fMRI) could provide insights into neural substrates of growth mindset and intrinsic motivation. We could measure the learner’s brain activity and neural responses to a specific task in relation to internal processes of motivation. For instance, intrinsic motivation could be assessed by an experimental task or free-choice behavior measures. 

The use of neuroscientific techniques enables us to focus on the learning process rather than the learning outcomes [29]. The neuroimaging findings offer an understanding of the brain, indicating the specific areas of brain activation which could in turn correlate with the behavioral results. As such, neuroimaging findings might support the self-reported data and explore brain regions with neural activation in relation to changes in performance during an online activity.

## 5. Neural Correlates of Growth Mindset and Intrinsic Motivation

There is a small body of existing growth mindset studies using neuroscience methods. The study by Moser et al. [30] suggested that individuals with a growth mindset are receptive to corrective feedback, exhibiting a higher Pe (error positivity) waveform response, which is correlated with a heightened awareness of and attention to mistakes. Enhanced Pe amplitude was associated with enhanced attention to corrective feedback following errors and subsequent error correction. Individuals with growth mindset are likely to have heightened awareness of and attention to errors. In addition, growth-minded individuals may neutralize the affective response to negative feedback, which could be indicated by neural activation. Anterior cingulate cortex (ACC) is the region of frontal midline cortex that is related to learning and control [31]. A recent study [32] found that growth mindset was related to both ventral and dorsal striatal connectivity with dorsal ACC. Dorsal ACC and dorsolateral prefrontal cortex (DLPFC) are critical to error-monitoring and behavioral adaptation. Growth mindset was strongly associated with dorsal and ventral striatal connectivity, as well as DLPFC. Learners with growth mindset are efficient in error-monitoring and receptive to corrective feedback. Hence, growth mindset has the potential to encourage intrinsically motivated behaviors in schools and promote lifelong learning.

Neuroscientific evidence has shown that ACC is associated to cognitive control and motivation [31]. Neural correlates revealed that dopamine is critical for motivation and cognitive control, with motivation-cognition interactions between midbrain regions and lateral frontal cortex [33]. Cognitive control is influenced by reward motivation. Participants were assigned to three levels of cognitive controls (low, mid and high). Different beneficial effects of reward (high versus low) were exhibited. Participants with high versus low reward anticipation showed increased activity in the medial and lateral frontal cortex. Brain activity was also stronger at the low level of cognitive control than mid and high levels. These findings demonstrated that motivation plays an important role in the cognitive control. In addition, high-level control tasks may demand an enhancing effect of motivation. 

A recent EEG study [34] showed that school children with growth mindset endorsement performed with higher accuracy after mistakes (i.e., post-error accuracy). The event-related potential (ERP), which is a measure of brain response due to the result of error and correct trials, revealed that Pe amplitude difference was largest at site Pz (i.e., midline parietal). Together with the behavioral data, correlational analyses showed that having a higher growth mindset was associated with a larger Pe difference. Students with attentional resources are able to remember their mistakes and able to make sense of their mistakes, thus correcting themselves during the learning process. Students do not like to take risks that show their weaknesses, such as making mistakes [35]. However, with growth mindset endorsement, students are not afraid to make mistakes, as they have the ability to learn with post-error accuracy. Hence, growth-minded students will be resilient and self-regulated when faced with obstacles or challenges during their learning process.

Little is known about the interplay between neural responses and intrinsic motivation. Intrinsically motivated action can be characterized by an individual’s engagement in behavior for one’s own sake, with free-choice time on a task [36]. An empirical study measured intrinsic motivation by examining a network of brain regions as the participants spent free-choice time on a word problem task [37]. Using fMRI, a network of brain regions revealed diminished task-related activity, predicting subsequent increased intrinsic motivation. The neuroimaging data suggest that decreased activation of neural cognitive control is associated with increased intrinsic motivation, thus extending one’s task engagement. Another recent study by Lee and Reeve [38] examined the neural substrates of intrinsic motivation during task performance. Their findings showed activated anterior insular cortex (AIC; a limbic-related cortex region) when students performed intrinsically motivated tasks. These neural findings are consistent with the concept of intrinsic motivation in terms of pursuit and interest satisfaction as intrinsic rewards. Based on these findings, it was concluded that AIC activity and its functional interactions are linked to an intrinsic-motivation neural system [38]. 

Two recent motivation studies used free-choice measures, such as a stop-watch (SW) game, as an experimental task to assess participants’ intrinsic motivation [39,40]. A traditional SW game includes a stopwatch that starts automatically, and the player tries to stop the watch at a specific time. Experimental stimuli were presented on the computer screen and participants were required to use the keypad to complete the SW tasks. It is interesting to note the relationship between the optimal challenge condition and intrinsic motivation using EEG [39]. Students performed better when they felt optimally challenged, and had enhanced intrinsic motivation in the game experiment. Stimulus-preceding negativity (SPN) is considered to be an electrophysiological indicator of motivation level. The EEG findings showed a larger SPN during the feedback anticipation period of the near miss condition than in the complete defeat condition, suggesting that participants were more intrinsically motivated to win in close games [39]. For the second study, fMRI was used to explore the degree of enjoyment for the preference levels of SW game [40]. It was found that participants had enhanced intrinsic motivation when they played the SW game with the action-outcome contingency condition. The fMRI findings revealed significant activation in the regions of the mid brain and ventral striatum in the action-outcome contingency condition, indicating that the intrinsic value of an action and achieving success. These two studies suggest that neuroscience methods are used to assess individuals’ intrinsic motivation using a free-choice experiment, such as a SW game. However, using game elements and design may have implications for authentic learning programs. Using the game approach, students may have enhanced intrinsic motivation for doing the activities in a gaming format or platform. Adopting the game approach and translating such motivation-enhancing elements into classrooms may seem challenging and time-consuming. Such experimental tasks are usually carried out in a closed environment, such as in a controlled laboratory setting within the fMRI facility. 

Intrinsic motivation is associated with sensitivity of feedback processing in the striatum [41]. The striatum plays a key role in reinforcing learning as it receives input from midbrain dopamine neurons and produces adaptive behaviors. Striatum activity is associated with reward processing, indicating that an intrinsically motivated task could foster the individual’s intrinsic motivation. For instance, feedback-related responses in the striatum can potentially promote or undermine intrinsic motivation of a desired behavior. Positive feedback was viewed as a rewarding outcome, and highly motivated subjects could attune to the feedback despite of fatigue through the study [41]. Performance-feedback may have affective salient response to striatum and produce a motivated behavior. A study by Lee [42] showed that intrinsic motivation was related to the AIC that is known to be associated with the sense of agency, while extrinsic motivation was associated with posterior parietal regions (e.g., posterior cingulate cortex, angular gyrus). The type of task also plays a very important role in activating the AIC. Lee [42] also found that interesting tasks activated the AIC and ventral striatum (i.e., brain region for reward processing), but not uninteresting tasks. AIC relates to the satisfaction of intrinsic need, whereas ventral striatum relates to the feeling of reward. His findings suggest that AIC and ventral striatum activations are associated with intrinsic motivation.

Intrinsic motivation is difficult to measure in an objective manner. In order to track one’s intrinsic motivation, it requires one to perform an experimental task over time. For instance, one’s brain activity can be tracked during the process of performing an intrinsically motivated or optimally challenged task. Together with behavioral measures, contemporary methods such as fMRI can be used to track the changes in intrinsic motivation during a free-choice activity.

## 6. The Neuroscientific Interplay between Growth Mindset and Intrinsic Motivation

Based on the abovementioned empirical findings, there is a distinctive neuroscientific interplay between growth mindset and intrinsic motivation. EEG findings could not directly show the brain regions that are related to mindset and motivation. Compared to the EEG, which is based on brain waveforms, fMRI is a better method for showing insights into the brain regions that are associated with growth mindset and intrinsic motivation. It is interesting to note that growth mindset is mainly associated with the dorsal regions of the brain, whereas intrinsic motivation is associated with the mid-brain regions. The common brain areas that are related to both growth mindset and intrinsic motivation are ACC and ventral striatum. Knowing the behavioral correlates for these two brain regions, potential research could investigate the neural correlates of growth mindset and intrinsic motivation. This brings us a step closer to understand the neural mechanism between growth mindset and intrinsic motivation. Below is a table that highlights the neuroscientific evidence of growth mindset and intrinsic motivation in relation to cognition. The behavioral correlate for the brain region is included in parentheses (see Table 1).

Growth mindset relates to brain processes, and brain processes relate to motivated behaviors. Likewise, motivated behaviors can affect cognition as motivation shapes what and how people think [43]. As such, individuals’ goals and needs may be exemplified when they steer their thinking towards desired outcomes. Research has shown that growth mindset has an impact on children’s behavior, particularly in terms of effort, motivation and resilience [12,44]. By understanding the underlying mechanism of intrinsic motivation, teachers are able to guide students in applying the relevant self-regulatory strategies at school. When individuals have intrinsic motivation for performing a task at work or school, their work or educational performance will improve [45,46]. With the inculcation of growth mindset, individuals will perceive the intrinsic value of a given task and self-regulate their behaviors to perform the task. Through internalization, individuals will generate intrinsically motivated behaviors at work or school.

As our brain is plastic, it is able to undergo reorganization and development. Brain plasticity or neuroplasticity refers to the ability of our brain to change throughout our life. It is thereby important to understand how our brain changes if we undergo growth mindset intervention and whether there are changes in our intrinsic motivation as well. This phenomenon is yet to be explored in educational research. It is thus an avenue worth pursuing for educators who hope to make the best of their students with regard to learning and personal growth. Such educational neuroscience research may impact teaching and learning, thus providing a better understanding of the neuroscientific interplay between growth mindset and intrinsic motivation. Future educational neuroscience research may include classroom interventions such as a growth mindset induction and how it affects the neuroscience of intrinsic motivation.

## 7. Future Directions

The principal intent of this paper is to highlight a potential educational neuroscience research in areas of growth mindset and intrinsic motivation. Although there are some empirical studies on mindset and motivation, the neuroscience of intrinsic motivation is still unclear and at its infancy. There are also limited neuroscientific studies on students’ motivation and learning. As educational neuroscience research looks promising in the near future, we should be aware of the potential integration between neuroscience methods and behavioral measures. For successful intervention studies, there are some considerations that need to be warranted. 

First, educators should design a task that has intrinsic value for students to be engaged in doing. For instance, an interesting task will instill curiosity into students, when compared to an uninteresting one. Inculcating the value of doing the task or task value will definitely stimulate the students’ interest. Second, teachers should provide the autonomy or choice for students. Autonomy or the agency of learning is the key substrate to intrinsic motivation [17]. Research has shown that autonomy is the strongest predictor of intrinsic motivation [47]. Autonomy is considered the self-endorsement of actions, whereby individuals feel less coerced and they generate autonomous behavior at work or school. In the same vein, choice is the opportunity for individuals to decide and exert control over the situation. A recent study found that the provision of choice, however trivial or inconsequential, might also increase an individual’s intrinsic motivation [48]. The researchers used behavioral and electrophysiological (i.e., electroencephalogram) evidence to explain the importance of need satisfaction for autonomy to enhance one’s intrinsic motivation toward the task. 

Third and finally, performance-related feedback could influence intrinsic motivation [41]. Participants are likely to perceive their performance on the task differently based on the type of performance-related feedback. For instance, positive feedback may enhance one’s intrinsic motivation, while negative feedback may undermine one’s motivation. In addition, the frequency of performance-related feedback may affect one’s neural processing (i.e., posterior cingulate cortex) in supporting task performance. There were enhanced activity of posterior cingulate cortex and performance gains after the performance-feedback manipulation. This shows that posterior cingulate cortex might facilitate the learning of a task [41]. The current level of a learner’s intrinsic motivation may also influence the way he or she processes the performance-related feedback. It is still not fully clear how the nature of performance-feedback could affect an individual’s feedback processing. Perhaps neuroscience methods could provide some insights into this area of research.

Based on the neuroscientific evidence, there is an undermining effect of monetary reward on intrinsic motivation; that is, one’s intrinsic motivation is undermined when extrinsic reward is no longer promised [26]. Neuroscience findings suggest that there are connections between the striatum and the prefrontal cortex in determining the outcome; decreased activation of the striatum and midbrain when the subjects do not get the task value, as well as decreased activation of the lateral prefrontal cortex (LPFC) when they are not motivated to show cognitive engagement with the task. Since growth mindset is a belief system that favors hard work and performance monitoring [32], a learner’s subjective belief in determining the outcome may modulate activity of the striatum, in response to cognitive feedback that nurtures growth mindset. Hence, neuroscientific evidence may provide insights into the learning and motivational processes that could be helpful for teachers and practitioners in improving their learning and teaching practices, thus supporting student learning and motivation. 

## 8. Concluding Remarks

This paper reviewed the recent empirical neuroscientific studies on growth mindset and intrinsic motivation. Research in these areas is still in its infancy. This paper attempted to provide an overview of the underlying mechanism between growth mindset and intrinsic motivation. Educating students about growth mindset and how they can improve their learning experience is a step toward increased intrinsic motivation in our society. From a personal perspective, intrinsic motivation is the key substrate to learning and development. The promotion of a growth mindset can nurture individuals to learn as they understand that intelligence is malleable. It is important that, as teachers, we show our students the value and importance of learning at schools. With a growth mindset, students will learn with a positive attitude, and they will identify the importance of the contents. Teachers should also embrace a growth mindset such that they will understand the importance of providing autonomy over student learning to enhance self-regulation. As such, students will be more motivated to learn subjects at school, rather than relying on the presumption that students will be interested in learning. This preliminary review paper offers a useful road map for identifying the areas that need to be addressed in neuroscientific research related to growth mindset and intrinsic motivation. However, this paper did not discuss the potential roles of socio-demographic variables and personality traits on growth mindset and intrinsic motivation. Future research will benefit from the continued development of neuroscientific evidence to connect the substantial behavioral evidence of these variables and traits associated with growth mindset and intrinsic motivation. 

## Figures and Tables

**Table 1 brainsci-08-00020-t001:** Neuroscientific evidence of growth mindset and intrinsic motivation.

Growth Mindset (Behavior)	Intrinsic Motivation (Behavior)
Enhanced Pe amplitude (awareness and attention) [30,34]	Enhanced SPN (engagement and enjoyment) [39]
DLPFC (error-monitoring and behavioral adaptation) [32]	Medial and lateral frontal cortex (cognitive control) [33]
Dorsal ACC (error-monitoring and behavioral adaptation) [32]	ACC (error-monitoring and behavioral adaptation) [37]
-	AIC (awareness, engagement) [38,42]
Dorsal and ventral striatum (intrinsic value of an action) [32]	Ventral striatum (intrinsic value of an action, reward processing) [40,41,42]

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
