# Peer review of "The Neuroscience of Growth Mindset and Intrinsic Motivation"

_brainsci, 2018, doi:10.3390/brainsci8020020_

Round 1

Reviewer 1 Report

General Comments

This manuscript provides a commendable case for the investigation of the neural basis of growth mindset and intrinsic motivation, and the author provides a review of some preliminary findings in that area.  As mentioned below, however, the review is somewhat cursory.  Thus, a question is whether the author intends for this paper to be a thorough review or simply a suggestion that the investigation of these topics is important to do.   Regardless, the manuscript is not ready for publication in its current form.

Assuming that the author restricts herself to the references that she has cited, it would be helpful to see a more thorough analysis and organization of the information.  There are some profound issues that are ignored or under-analyzed.  For example, there are broad statements like: “ Intrinsic motivation is associated with dopaminergic value system [25, 26].”  In fact, the Murayama paper is about the undermining of motivation by extrinsic rewards – a much more subtle and intricate issue than the author’s statement implies.  Later (line 204) there is a similar statement: “Intrinsic motivation is associated with the striatum [44].” Yes, but is the activity unique to intrinsic motivation or does it relate to all positive experiences? Discussion of issues like this would add depth to the manuscript.

On line 126, the author refers to an “anterior prefrontal P3 response,” but the cited paper refers to the Pe waveform.  It would be helpful to explain the relationship between these.

On lines 164-175, there is reference to two papers without any explanation of the neural findings.  A similar problem applies to lines 247-251.

Specific Wording Issues

There are several places where the text doesn’t make sense or needs some correction:

On line 61, who is “we”?

On line 67, there should not be any comma.

On line 68, the words should be “junior high school students.”

On line 75, the sentence needs to be reworded.

On lines 86 and 87, the word “innate” is used incorrectly.

On line 114, what is “this area of research?”

On line 119, “better” than what?

On line 129 there is reference to a “recent study” with no citation.

On line 143, “reminiscent” of what?

On line 152, who is “they?”

On line 156, “region” should be inside the parentheses

On line 137, the statement is a non sequitur from the rest of the paragraph: “Although cognitive control is influenced by reward motivation, growth 
mindset has the potential to encourage intrinsically motivated behaviors in schools and promote 
lifelong learning. 
”

On line 160, there is a statement that does not reflect the cited article: “Striatum activity is associated with reward processing, indicating that intrinsically motivated task 
could foster the individual’s intrinsic motivation. 
” First, the sentence does not make sense.  Second, the striatum is not mentioned in the Lee and Reeve article; the author also cites Lee’s dissertation later on, and there it is mentioned with respect to a different experiment.

On line 197, the sentence does not make sense.

On line 240, this sentence needs to be reworded: “The task to be designed should be intrinsically 
motivated or task-specific motivation. 
”

On line 259 the DePasque reference is in the wrong format.

I hope that the author finds these comments helpful.

Author Response

Reviewer 1 comments

This manuscript provides a commendable case for the investigation of the neural basis of growth mindset and intrinsic motivation, and the author provides a review of some preliminary findings in that area.  As mentioned below, however, the review is somewhat cursory.  Thus, a question is whether the author intends for this paper to be a thorough review or simply a suggestion that the investigation of these topics is important to do.   Regardless, the manuscript is not ready for publication in its current form.

Thanks for your review and giving me constructive feedback to improve on the manuscript. Really appreciate your time and suggestions. Please see my responses in red to your comments/ suggestions below. Thanks.

Assuming that the author restricts herself to the references that she has cited, it would be helpful to see a more thorough analysis and organization of the information.  There are some profound issues that are ignored or under-analyzed.  For example, there are broad statements like: “Intrinsic motivation is associated with dopaminergic value system [25, 26].”  In fact, the Murayama paper is about the undermining of motivation by extrinsic rewards – a much more subtle and intricate issue than the author’s statement implies.  Later (line 204) there is a similar statement: “Intrinsic motivation is associated with the striatum [44].” Yes, but is the activity unique to intrinsic motivation or does it relate to all positive experiences? Discussion of issues like this would add depth to the manuscript.

Yes, thanks for the comments. Had amended the broad statements.

Please see the red text, lines 113-116 and line 230. Please take note that the reference number has changed due to new reference. Reference number 44 has changed to #45.

On line 126, the author refers to an “anterior prefrontal P3 response,” but the cited paper refers to the Pe waveform.  It would be helpful to explain the relationship between these.

Sorry for the overlook here. Had amended to the correct term. Please see the red text, lines 142-143.

On lines 164-175, there is reference to two papers without any explanation of the neural findings.  A similar problem applies to lines 247-251.

Sorry for missing the explanation of the neural findings. Had inserted. Please see the red text, lines 143-146 and lines 149-154.

Specific Wording Issues

There are several places where the text doesn’t make sense or needs some correction:

On line 61, who is “we”?

Amended and please see the red text, line 75.

On line 67, there should not be any comma.

Amended and please see the red text, line 82.

On line 68, the words should be “junior high school students.”

Amended and please see the red text, line 83.

On line 75, the sentence needs to be reworded.

Amended and please see the red text, lines 90-91.

On lines 86 and 87, the word “innate” is used incorrectly.

Changed to “inherent”. Please see the red text, line 99.

On line 114, what is “this area of research?”

Changed to “such neuroscientific research”. Please see the red text, line 130.

On line 119, “better” than what?

Deleted the word “better”. Please see line 135.

On line 129 there is reference to a “recent study” with no citation.

The recent study is referring to Ref #32 that is placed at the end of the statement. Shifted it to the beginning of statement. Please see the red text, line 148.

On line 143, “reminiscent” of what?

Amended and please see the red text, line 165.

On line 152, who is “they?”

 Amended to “The neuroimaging”. Please see the red text, line 175.

On line 156, “region” should be inside the parentheses

Amended. Please see the red text, line 179.

On line 137, the statement is a non sequitur from the rest of the paragraph: “Although cognitive control is influenced by reward motivation, growth 
mindset has the potential to encourage intrinsically motivated behaviors in schools and promote lifelong learning.”

Thanks for the highlight. Removed this statement. Please see line 161.

On line 160, there is a statement that does not reflect the cited article: “Striatum activity is associated with reward processing, indicating that intrinsically motivated task 
could foster the individual’s intrinsic motivation. ” First, the sentence does not make sense.  Second, the striatum is not mentioned in the Lee and Reeve article; the author also cites Lee’s dissertation later on, and there it is mentioned with respect to a different experiment.

Sorry for the overlook. Removed the statement. Please see the red text, line 182.

On line 197, the sentence does not make sense.

Sorry for the overlook. Amended the statement. Please see the red text, line 230.

On line 240, this sentence needs to be reworded: “The task to be designed should be intrinsically motivated or task-specific motivation. 
”

Thanks for the suggestion. Amended and please see the red text, line 269.

On line 259 the DePasque reference is in the wrong format.

Sorry for the overlook. Amended and please see line 288.

Reviewer 2 Report

This manuscript aimed to investigate whether the growth mindset and intrinsic motivation improves learning, especilly in undergraduate research students. While the hypotheses from this study could potentially be interesting and important, it would be necessary to complete with additional papers in order to offer a good background to support the above mentioned link. Thus, there are several problems with this manuscript, which I discuss below. 

Major limitations:

1) The “introduction” is a bit short. In fact, it would be recommendable to provide more insight regarding the importance of this study. Additionally, it would be nice to describe the following parts of the manuscript.

2) Please state the inclusion/exclusion criteria explicitly for this study.

3) I strongly desagree with this kind of sentence: “Dopamine is a neurotransmitter that aids in controlling the brain’s reward and pleasure centres.” It is difficult to explain behavior or mental processess with a single neurotransmitter. In fact, this kind of sentences seems a litte bit reduccionist. Hence, it would be interesting to discuss this part with additional neurotransmitters, neurohormones and their interactions.

4) They did not control the potential role of socio-demographical variables in his/her model. Due to the fact that your study is focused in a specific link in a heterogeneous sample, you should control the effect of these socio-demographic variables.

5) It should be recommendable to provide additional research to justify his/her hypothesis. In fact, he/she need more neuroscientific papers.

6) I would recommend to include as limitations: personality traits, students interests, absence of a considerable research in this field (neuroscientific results,   etc.

Author Response

Reviewer 2

This manuscript aimed to investigate whether the growth mindset and intrinsic motivation improves learning, especially in undergraduate research students. While the hypotheses from this study could potentially be interesting and important, it would be necessary to complete with additional papers in order to offer a good background to support the above mentioned link. Thus, there are several problems with this manuscript, which I discuss below. 

Thanks for your review and giving me constructive feedback to improve on the manuscript. Really appreciate your time and suggestions. Please see my responses in red to your comments/ suggestions below. Thanks.

Major limitations:

1) The “introduction” is a bit short. In fact, it would be recommendable to provide more insight regarding the importance of this study. Additionally, it would be nice to describe the following parts of the manuscript.

Thanks for the suggestion. Inserted more details to the introduction. Please see the red text, lines 40-52.

2) Please state the inclusion/exclusion criteria explicitly for this study.

Thanks for the suggestion. This paper is a preliminary review paper to provide recent empirical studies on educational neuroscientific research related to growth mindset and intrinsic motivation. It is not a systematic review paper and that is why the inclusion/exclusion criteria are not stated.

3) I strongly disagree with this kind of sentence: “Dopamine is a neurotransmitter that aids in controlling the brain’s reward and pleasure centres.” It is difficult to explain behavior or mental processes with a single neurotransmitter. In fact, this kind of sentences seems a little bit reductionist. Hence, it would be interesting to discuss this part with additional neurotransmitters, neurohormones and their interactions.

Sorry for the overlook and amended. Kindly see the red text, lines 113-116.

4) They did not control the potential role of socio-demographical variables in his/her model. Due to the fact that your study is focused in a specific link in a heterogeneous sample, you should control the effect of these socio-demographic variables.

Thanks for your suggestion. However, the socio-demographics variables are not mentioned in most studies and it is difficult to discuss about this. This point is taken note and mentioned as a limitation in the paper. Please see red text, lines 319-323.

5) It should be recommendable to provide additional research to justify his/her hypothesis. In fact, he/she need more neuroscientific papers.

Thanks for your suggestion and amended. Kindly see the red text, lines 47-52 and 292-303.

6) I would recommend to include as limitations: personality traits, students’ interests, absence of a considerable research in this field (neuroscientific results,   etc.

Thanks for your suggestion. Inserted and please see the red text, lines 319-323.

Round 2

Reviewer 1 Report

The author has done a very responsible job of making adjustments in the manuscript, and I applaud her effort to address the items that I pointed out.  Although the manuscript is improved, the organization and coverage still do not provide the reader with a clear picture of the current state of the literature.  Instead of a list of research studies, I would encourage the author to organize the paper according to the major informative findings that provide an understanding of the processes at hand.

I’m still not absolutely clear about the relationship between growth mindset and intrinsic motivation.  The new language (lines 40-52) attempts to address the relationship between the two concepts, but there still remains the dilemma that they are on different dimensions, one being an attitude and the other a motivation. Are they always coexistent?  Can extrinsic motivation contribute to growth mindset under certain circumstances?  If we look at studies of intrinsic motivation, can we always be confident that they will tell us about growth mindset?  Even after reading sections 3 and 4 of the manuscript, I am unclear about the answers to these questions, and until the relationship of growth mindset and intrinsic motivation are firmly established, the neural findings discussed in sections 5 and 6 are fuzzy in terms of how they relate to both concepts.

Specific Conceptual or Wording Issues

Line 41: “Though these …” I don’t understand this sentence.

Line 89: … outcomes in particular subjects …

Line 100: “This innate energy …” What is this referring to from the previous sentence? Motivation, behavior, or determination? How do we know it is innate?

Line 109: subjects and to differentiate their interests

Line 115: … rewarding events project …

Line 117: “Dopamine 
is considered a key substrate of intrinsic motivation, thus promoting attentiveness and behavioral 
engagement [26, 27]. 
”  Those references are not the appropriate papers to cite in supporting the statement, and the rest of that paragraph needs a substantial rewrite to provide a coherent explanation.

Line 139: A better title would be: Neural correlates of growth mindset and intrinsic motivation.

Line 142: …response that is correlated with a heightened …

Line 154: Reference 32 does not mention P3, except when reviewing another study. 

Line 159: Participants were assigned to three levels of cognitive control (low …

Line 160-161: This sentence doesn’t make sense without rewording it.

Lines 163-165: This sentence doesn’t make sense without rewording it.

Lines 162-169: What were the neural findings of the study?

Line 183: A description of the stop-watch game would be helpful. Without that explanation, the next sentences do not make sense.

Line 202: within the fMRI …

Line 209: educators seeking to positively impact …

Line 238: A study by Lee [46] showed that intrinsic …

Line 269: sentence needs rewording

Line 287-290: I don’t understand these sentences

Line 299: may modulate activity of the anterior striatum

The date for Reference 27 is listed in Pubmed as 2015 and not 2013.

Author Response

Hi Reviewer 

Great thanks to your comments and feedback.

Really appreciate your time and effort.

Thanks for helping me to improve the manuscript.

Please see my responses to your comments in the attached pdf file.

Thank you very much.

Reviewer 2 Report

Authors have successfully answered my comments.

Author Response

Hi Reviewer

Great thanks for approving my responses to the first round of review.

Thank you very much.